# Probiotic Formulation VSL#3 Interacts with Mesenchymal Stromal Cells To Protect Dopaminergic Neurons via Centrally and Peripherally Suppressing NOD-Like Receptor Protein 3 Inflammasome-Mediated Inflammation in Parkinson's Disease Mice

Liping Zhou,[a] Deqiang Han,[a] Xingzhe Wang,[a] Zhiguo Chen[a,b,c]

aCell Therapy Center, Beijing Institute of Geriatrics, Xuanwu Hospital Capital Medical University, National Clinical Research Center for Geriatric Diseases, Key Laboratory of Neurodegenerative Diseases, Ministry of Education, Beijing, China
bCenter of Neural Injury and Repair, Beijing Institute for Brain Disorders, Beijing, China
cCenter of Parkinson's Disease, Beijing Institute for Brain Disorders, Beijing, China

**ABSTRACT** Systemic immunomodulation is increasingly recognized among the beneficial effects of mesenchymal stromal cells (MSCs) in treatment of Parkinson's disease (PD), while the underlying mechanism is not fully understood. With the growing popularity of using probiotics as an adjuvant approach in PD treatment, concerns about the added effects of probiotics have been raised. In addition to the molecular mechanism mediating the neuroprotective effects of MSCs, the combined effects of a probiotic formulation, VSL#3, and MSC infusion were also evaluated in PD mice. The animals were weekly treated with human MSCs (hMSCs) via the tail vein, VSL#3 via the gastrointestinal tract, or their combination six times. hMSCs, VSL#3 alone, and their combination markedly ameliorated the decreased striatal dopamine content, loss of dopaminergic neurons in the substantia nigra, increased levels of proinflammatory cytokines in serum, as well as tumor necrosis factor alpha (TNF-$\alpha$) and interleukin-1$\beta$ (IL-1$\beta$) mRNAs in striatum and peripheral tissues induced by MPTP. Furthermore, hMSCs, VSL#3, and their combination notably downregulated mRNA expression of NOD-like receptor protein 3 (NLRP3) and caspase-1 in brain and peripheral tissues of PD mice. These results suggest that hMSCs, VSL#3, and their combination prevent neurodegenerative changes in PD mice via anti-inflammatory activities in both the central and peripheral systems, possibly through suppressing the NLRP3 inflammasome. Moreover, two-way analysis of variance (ANOVA) indicated that VSL#3 interacts with hMSCs to attenuate neurodegeneration and inhibit NLRP3 inflammasome-mediated inflammation without altering the effects of hMSCs. Major findings of our study support the usage of probiotic formulation VSL#3 as an adjuvant therapy to hMSC infusion in PD treatment.

**IMPORTANCE** This study provides evidence for the neuroprotective activities of human umbilical cord MSCs from the aspect of anti-inflammation actions. hMSCs inhibit the NLRP3 inflammasome and MPTP-induced inflammation in both brain and periphery to relieve the degenerative changes in dopaminergic neurons in PD mice. Furthermore, as an additional therapeutic agent, probiotic formulation VSL#3 interacts with hMSCs in suppressing the NLRP3 inflammasome as well as the central and peripheral anti-inflammatory effects to exert neuroprotective actions in PD mice without altering the actions of hMSCs, suggesting the potential of VSL#3 as an adjuvant therapy in PD treatment. The findings of the present study give a further understanding of the anti-inflammatory activity and the molecular mechanism for the beneficial effects of MSCs as well as the potential application of probiotic formulation as an adjuvant approach to MSC therapy in PD treatment.

Address correspondence to Zhiguo Chen, chenzhiguo@gmail.com.

The authors declare no conflict of interest.

10.1128/spectrum.03208-22 **1**

**KEYWORDS** mesenchymal stromal cells, probiotic, Parkinson's disease, anti-inflammatory, NLRP3 inflammasome

Parkinson's disease (PD) is the second most frequent neurodegenerative diseases after Alzheimer's disease (AD). The prevalence of PD increases with aging, especially among elderly (1). The typical pathological change of PD is a selective loss of dopamine-producing neurons in the substantia nigra with a consequent drop in dopamine production and degeneration of the nigrostriatal pathway (2), inducing a number of motor and nonmotor symptoms. Current therapeutic strategies, including pharmacotherapy and deep brain stimulation (DBS), predominantly target the symptoms of PD and have shown some benefits (3, 4). However, because of the unclear pathophysiology underlying PD, there is no cure for PD. Recently, inflammation has been increasingly identified as a bridge between genetic susceptibility and environmental factors copromoting PD (5). The crucial role of neuroinflammation in the pathogenesis of PD has been proven in human trial and preclinical models; neuroinflammation may further drive the progressive loss of dopaminergic neurons. Moreover, a deleterious role of peripheral inflammation has been recently recognized in sensitizing microglia-associated inflammation in PD (6, 7). Therefore, increasing efforts on anti-inflammation approaches are being made in developing a cure for PD.

Stem cell-based therapy is an attractive approach for treatment of PD due to their potential for replacing degenerated neurons. Among stem cells, mesenchymal stromal cells (MSCs) have gained the most attention because of their capacity for self-renewal, easy isolation, and freedom from ethical problems. Possessing the potential to differentiate toward neuron-like cells, MSCs were initially thought to be effective in PD treatment via replacing the degenerated neurons (8). However, the fact that systemic transplantation of MSCs via intravenous and intranasal injection also protected dopaminergic neurons (9, 10) and significantly alleviated locomotor deficits (11) in animals with 1-methyl-4-phenyl-1,2,3,6-tetrahydropyridine (MPTP)-induced PD suggests that MSCs may not necessarily counteract PD through direct cell replacement and systemic mechanisms must be involved in mediating the neuroprotective actions of MSCs. Actually, recent studies have shown that MSCs exert immunomodulatory, anti-inflammatory, neurotrophic, and angiogenetic effects to create a favorable microenvironment for neural regeneration. Concerning the anti-inflammatory function, MSCs secrete a variety of soluble molecules and modulate the cytokine production of the host (12). However, the mechanism by which MSCs exert anti-inflammatory activity remains unclear.

The NOD-like receptor protein 3 (NLRP3) inflammasome is reported as the most clinically affected inflammasome. It is a key innate immune sensor, and the activation of its main effector, caspase-1, promotes the secretion and maturation of pro-interleukin-1$\beta$ (pro-IL-1$\beta$) to produce active IL-1$\beta$, which could, in turn, trigger the inflammatory responses. This function is crucial for the modulation of neuroinflammation mediated by microglia (13). Recent evidence particularly suggests the critical importance of NLRP3 inflammasome in PD progression via caspase-1 activation. In human postmortem brain from PD cases, NLRP3 expression was found to be elevated in mesencephalic neurons; additionally, NLRP3 genetic polymorphisms were associated with reduced NLRP3 activity and PD risk (14). These findings suggest that the precise modulation of the NLRP3 inflammasome might be a novel therapeutic strategy in PD. Inhibitors of the NLRP3 inflammasome indeed point to a potential augmentation in PD treatment (15).

Meanwhile, gastrointestinal dysfunction is frequently identified in PD patients and is reported to be a contributor to PD pathogenesis (16). Research has suggested that symptoms of PD may be related to gut dysbiosis and gut barrier dysfunction that results in inflammation. PD patients show a different gut microbial composition from that of healthy individuals, and this dysbiosis is more evident in the severe PD phenotype (17). Based on these findings, target modulation of the gut microbiome might be an alternative approach for management of PD. VSL#3 is a probiotic formulation consisting of eight cultured bacteria that is currently prescribed for treatment of irritable bowel syndrome via modulating intestinal permeability. Recent studies found that VSL#3 improved intestinal inflammation and microbial imbalance

in mice (18). Further study demonstrated the neuroprotective effect of VSL#3 on dopaminergic neurons in a model of PD via dampening the inflammation and improving neuronal performance (19), suggesting that VSL#3 might be a potential candidate that could be implemented as an alternative approach for treatment of PD.

With regard to the regulation of infused MSCs on the gut microbiota reported in a rat model of ischemic stroke (20), concerns about the combined effects of administering a probiotic and MSCs have been raised. Based on these findings, the present study aimed to determine the added effect of administering VSL#3 as an adjuvant approach to MSC treatment in a mouse model of MPTP-induced PD as well as the NLRP3 inflammasome-related mechanism.

## RESULTS

**Characterization of human umbilical cord MSCs.** As in our previous study (21), the human MSCs (hMSCs) employed in the present study were positive for CD73, CD90, and CD105 but were negative for CD45 and HLA-DR.

**VSL#3 interacts with hMSCs to regulate the striatum contents of DA and NE but not DOPAC and HVA.** To investigate the changes in dopamine production in response to MPTP and treatments with VSL#3, hMSCs as well as their combination, the striatal contents of dopamine (DA), 3,4-dihydroxyphenylacetic acid (DOPAC), homovanillic acid (HVA), and norepinephrine (NE) were determined by high-performance liquid chromatography (HPLC) assay. Compared with the case with control mice, the content of DA and its metabolites decreased significantly in the striatum of MPTP-treated mice (Fig. 1, $P < 0.001$ versus control). VSL#3 and hMSCs alone significantly restored the change in DA content (Fig. 1A, $P < 0.05$ versus PD), while the increases in DOPAC, HVA, and NE did not reach statistical significance. Combination of VSL#3 and hMSCs induced noticeable elevations in DA, DOPAC, and HVA (Fig. 1A to C, $P < 0.05$ versus PD) without altering the content of NE in PD mice. Results of two-way analysis of variance (ANOVA) indicated that interactions between VSL#3 and hMSCs restored changes in DA (VSL#3 × hMSC interaction, $P = 0.0006$) and NE content (VSL#3 × hMSC interaction, $P = 0.0060$). The contents of DOPAC and HVA in the group receiving PD plus VSL#3 plus hMSCs (PDVM group) appeared to be higher than that in the groups receiving each alone. However, the differences were not statistically significant, and two-way ANOVA suggested no interaction between VSL#3 and hMSCs in regulating DOPAC and HVA.

**VSL#3 interacts with hMSCs to attenuate the loss of dopaminergic neurons in the SN in PD mice.** Besides decreased striatal production of DA, dopaminergic neuron damage is associated with the loss of tyrosine hydroxylase (TH)-positive cells in the substantia nigra (SN). Consistent with the changes in neurotransmitters, MPTP induced a >65% reduction in survival of the dopaminergic neurons relative to that in control mice (Fig. 2, $P < 0.001$ versus control). Notably, the loss of dopaminergic neurons in the substantia nigra in response to MPTP was significantly ameliorated by VSL#3 and hMSCs alone as well as the combined use of them, and the survival of TH-positive neurons increased to 53.6%, 55.6%, and 61.0% that of control mice, respectively (Fig. 2, $P < 0.01$ versus PD), demonstrating the beneficial effects on dopaminergic neurons. According to the results of two-way ANOVA, VSL#3 interacted with hMSCs in preventing the loss of dopaminergic neurons (VSL#3 × hMSC interaction, $P = 0.0041$), but they induced no alteration in the response of neurons to each of them.

**VSL#3 interacts with hMSCs to restore the mRNA levels of inflammatory cytokines in striatum in PD mice.** Neuroinflammation plays a crucial role in the pathogenesis of PD due to release of various proinflammatory cytokines. Real-time PCR revealed a >2-fold elevation in mRNA expression of tumor necrosis factor alpha (TNF-$\alpha$) and IL-1$\beta$ in the striatum of MPTP mice (Fig. 3A and B, $P < 0.05$ versus control). In view of the crucial role of the NLRP3 inflammasome for neuroinflammation and IL-1$\beta$ release (22), the mRNA expression levels of NLRP3 and caspase-1 were also measured. In the striatum of PD mice, both caspase-1 and NLRP3 mRNA levels were remarkably upregulated (Fig. 3C and D, $P < 0.05$ versus control). Administration with VSL#3 or hMSCs alone and their coadministration restored the mRNA expression of TNF-$\alpha$, IL-1$\beta$, caspase-1, and NLRP3 in striatum to a level comparable to that of control mice ($P < 0.05$ versus PD), confirming that both VSL#3 and hMSCs targeted the neuroinflammatory response in PD mice. Two-way ANOVA implied an interaction between

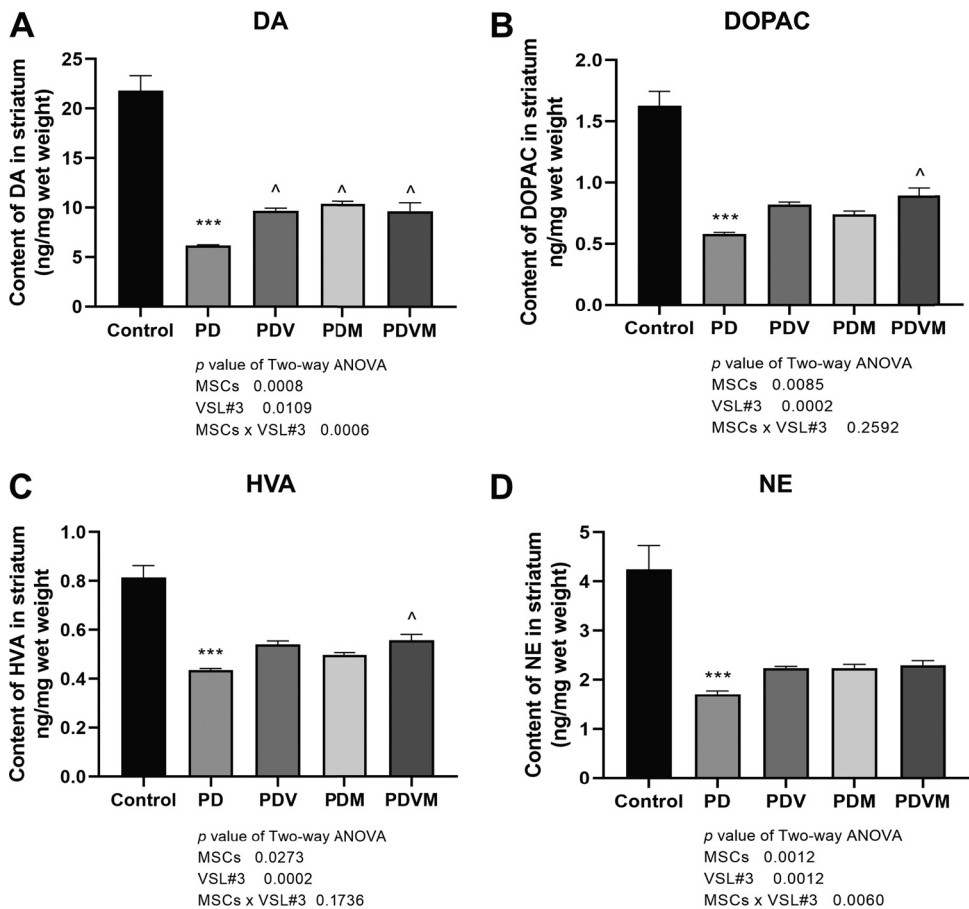

**FIG 1** Effects of VSL#3, human mesenchymal stromal cells (hMSCs), and their combination on striatum content of dopamine and its metabolites in PD mice. After administration, mice were euthanized and the left striatum was freshly collected and weighed. After homogenization, HPLC assay was performed to determine the contents of dopamine (DA) (A) and its metabolites 3,4-dihydroxyphenylacetic acid (DOPAC) (B), homovanillic acid (HVA) (C), and norepinephrine (NE) (D). Data are expressed as means ± SEM ($n = 6$). ***, $P < 0.001$ versus control; ^, $P < 0.05$ versus PD. MPTP, 1-methyl-4-phenyl-1,2,3,6-tetrahydropyridine; PD, Parkinson's disease group; PDV, Parkinson's disease group receiving VSL#3 treatment; PDM, Parkinson's disease group receiving MSC infusion; PDVM, Parkinson's disease group receiving cotreatment with VSL#3 and MSCs.

VSL#3 and hMSCs in inhibiting the neuroinflammatory responses in the striatum in PD mice (VSL#3 × hMSC interaction, $P = 0.0237$ for TNF-$\alpha$ mRNA, $P < 0.0001$ for IL-1$\beta$ mRNA, $P = 0.0030$ for caspase-1 mRNA, and $P < 0.0001$ for NLRP3 mRNA). VSL#3 did not alter the responses of TNF-$\alpha$, IL-1$\beta$, caspase-1, or NLRP3 mRNA expression to hMSC infusion, while hMSC notably enhanced the action of VSL#3 on inhibiting NLRP3 mRNA expression in the striatum ($P < 0.05$ versus PDV).

**VSL#3 interacts with hMSCs to suppress the serum level of inflammatory cytokines in PD mice.** Recently, a deleterious role of peripheral inflammation in promoting neuroinflammation in PD has become evident (6, 7). In order to investigate the effects of MPTP, VSL#3, and hMSCs alone and the combined application of them on systemic inflammation, serum levels of proinflammatory cytokines, including TNF-$\alpha$, IL-1$\beta$, IL-6, IL-17, granulocyte-macrophage colony-stimulating factor (GM-CSF), and gamma interferon (IFN-$\gamma$) were measured. MPTP injection induced a noticeable boost in serum levels of TNF-$\alpha$, IL-1$\beta$, IL-6, IL-17, GM-CSF, and IFN-$\gamma$: 2.33-fold, 4.53-fold, 4.57-fold, 12.28-fold, 5.56-fold, and 1.90-fold in PD mice, respectively (Fig. 4, $P < 0.01$ versus control); these levels were completely reversed by VSL#3 and hMSCs alone and in combination ($P < 0.05$ versus PD). Two-way ANOVA identified the interactions between VSL#3 and hMSCs in downregulating the circulating proinflammatory cytokines (VSL#3 × hMSC interaction, $P < 0.0001$ for TNF-$\alpha$, $P = 0.0006$ for IL-1$\beta$, $P = 0.0016$ for IL-6, $P = 0.0332$ for IL-17, $P = 0.0001$ for IFN-$\gamma$, and $P < 0.0040$ for GM-CSF).

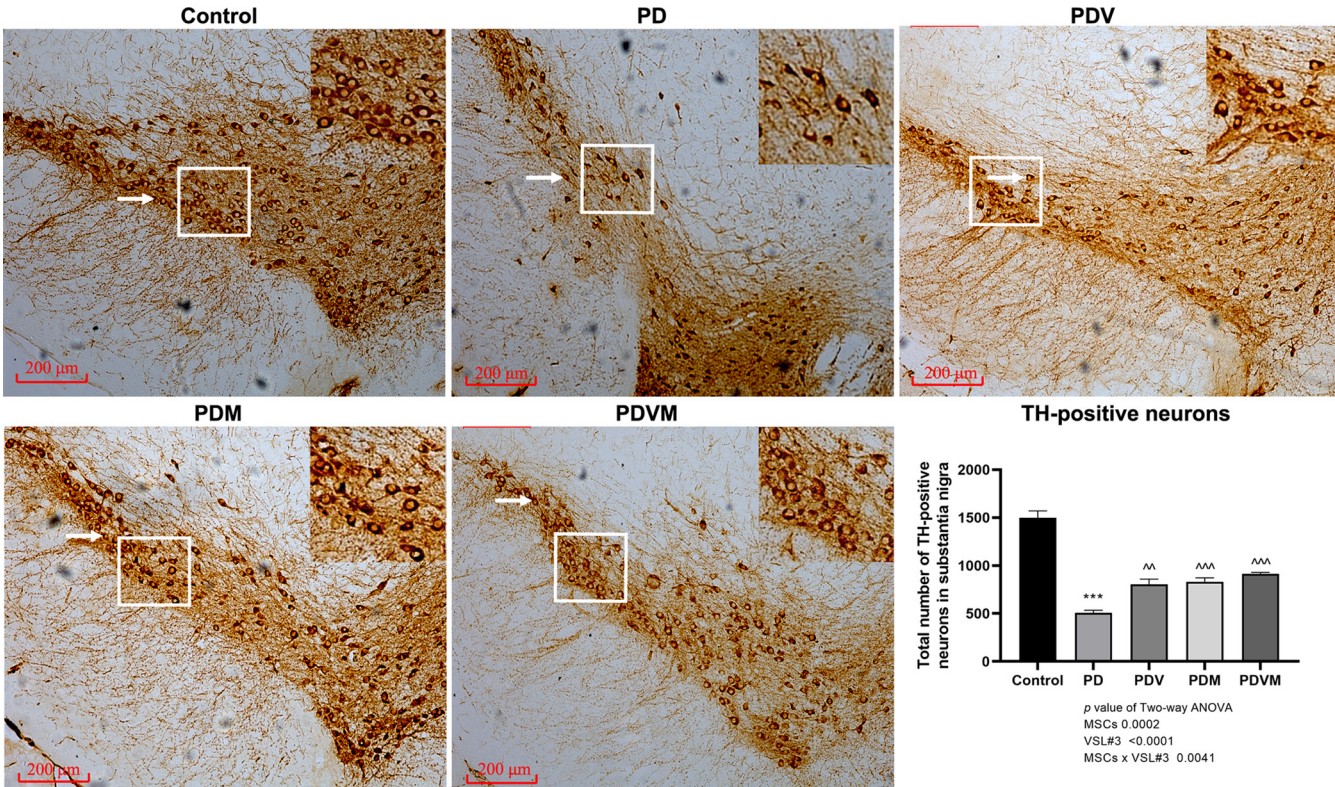

**FIG 2** Effects of VSL#3, hMSCs, and their combination on MPTP-induced tyrosine hydroxylase (TH)-positive neuronal loss in the substantia nigra of PD mice. At euthanasia, mice were perfused with 4% paraformaldehyde. The brain was isolated and collected. Immunohistochemistry was performed to visualize the TH-positive neurons (as indicated by the white arrow, 100×) in the substantia nigra of the control group, PD group, and PDM group. In the graph, data are expressed as means ± SEM ($n = 6$). ***, $P < 0.001$ versus control; ^^, $P < 0.01$ versus PD; ^^^, $P < 0.001$ versus PD.

However, either VSL#3 or hMSCs altered the peripheral inflammatory responses to each of them in PD mice.

**VSL#3 interacts with hMSCs to downregulate the mRNA expression of inflammatory cytokines in liver of PD mice.** Circulating inflammatory stimuli can get into the brain, stimulating synthesis of cytokines that, in turn, induce a peripheral inflammatory response (23). To investigate the involvement of peripheral inflammation in the pathogenesis of PD and response to MSC infusion, the expression level of cytokines was measured in the liver and intestine. Systemic injection of MPTP induced a striking upregulation in TNF-$\alpha$ and IL-1$\beta$ mRNA levels in the liver (Fig. 5A and B, $P < 0.05$ versus control), which were markedly suppressed by VSL#3, hMSCs, and cotreatment with them ($P < 0.05$ versus PD). Moreover, VSL#3 interacted with hMSCs in suppressing TNF-$\alpha$ and IL-1$\beta$ mRNA expression in the liver (VSL#3 $\times$ hMSC interaction, $P = 0.0336$ for TNF-$\alpha$ mRNA and $P < 0.0001$ for IL-1$\beta$ mRNA), but they did not induce alteration in the actions of either of them. Compared to the case with control mice, MPTP induced a noticeable increase in caspase-1 mRNA expression in liver (Fig. 5C, $P < 0.05$ versus control). Gene expression of NLRP3 increased in liver as well, while the difference did not reach statistical significance (Fig. 5D). VSL#3, hMSCs, and their combination restored the mRNA levels of caspase-1 and NLRP3 in liver to levels comparable to those of control mice ($P < 0.05$ versus PD). Two-way ANOVA indicated that VSL#3 interacted with hMSCs to exert an inhibitory effect on the NLRP3 mRNA level (VSL#3 $\times$ hMSC interaction, $P = 0.0058$) but not the caspase-1 mRNA level in striatum of PD mice (VSL#3 $\times$ hMSC interaction, $P = 0.4136$).

**VSL#3 interacts with hMSCs to downregulate the mRNA expression of inflammatory cytokines in intestine of PD mice.** The intestinal tissue of PD mice also showed outstanding increases in mRNA expression of TNF-$\alpha$ and IL-1$\beta$ (Fig. 6A and B, $P < 0.05$ versus control). The increased expression of TNF-$\alpha$ mRNA was notably downregulated by VSL#3 alone and the combined use of VSL#3 and hMSCs but not by hMSCs alone ($P < 0.05$ versus PD).

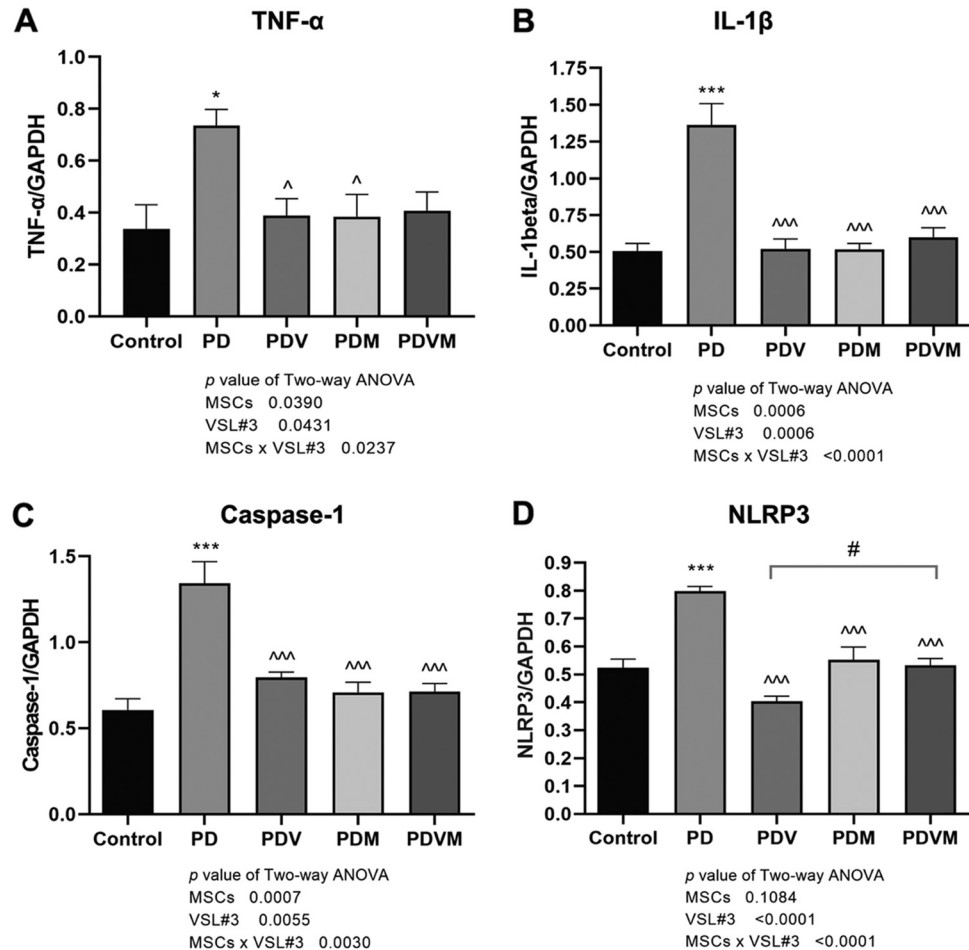

**FIG 3** Effects of VSL#3, hMSCs, and their combination on MPTP-induced mRNA expression of inflammatory cytokines and NLRP3 inflammasome molecules in the striatum of PD mice. At euthanasia, the right striatum was freshly collected. mRNA expression levels of TNF-$\alpha$, IL-1$\beta$, caspase-1, and NLRP3 were measured by real-time PCR. Data are expressed as means $\pm$ SEM ($n$ = 6). GAPDH, glyceraldehyde-3-phosphate dehydrogenase. *, $P < 0.05$ versus control; #, $P < 0.05$ versus PDV; ***, $P < 0.001$ versus control; ^^^, $P < 0.001$ versus PD.

VSL#3 and hMSCs alone and cotreatment with them inhibited NLRP3 mRNA expression as well, but the changes did not reach statistical significance. No interaction between VSL#3 and hMSCs was demonstrated by two-way ANOVA in regulating TNF-$\alpha$ and IL-1$\beta$ mRNAs in the intestinal tissue (VSL#3 $\times$ hMSC interaction, $P$ = 0.1501 for TNF-$\alpha$ mRNA and $P$ = 0.1029 for IL-1$\beta$ mRNA). In response to MPTP, caspase-1 and NLRP3 mRNA expression markedly increased in the intestinal tissue of PD mice (Fig. 6C and D, $P < 0.05$ versus control); levels were completely restored by VSL#3, hMSCs, and their combination to levels comparable to those in control mice ($P < 0.01$ versus PD). Two-way ANOVA showed that VSL#3 interacted with hMSCs in inhibiting caspase-1 and NLRP3 mRNA expression (VSL#3 $\times$ hMSC interaction, $P$ = 0.0020 for caspase-1 mRNA and $P$ = 0.0020 for NLRP3 mRNA). However, no difference was determined between VSL#3 or hMSCs alone and in combination.

## DISCUSSION

Studies demonstrate that mesenchymal stromal cells exert neuroprotective effects at least partially by the anti-inflammation-based immunomodulatory activities. The present study confirmed the anti-inflammatory and beneficial effects of human mesenchymal stromal cells (hMSCs) isolated from human umbilical cord in MPTP-induced PD mice model. Furthermore, we reported that hMSCs inhibited the proinflammatory cytokines and molecules of the NLRP3 inflammasome, a crucial inflammatory mechanism of PD progression and a novel therapeutic target of PD treatment, suggesting that the anti-inflammation-mediated

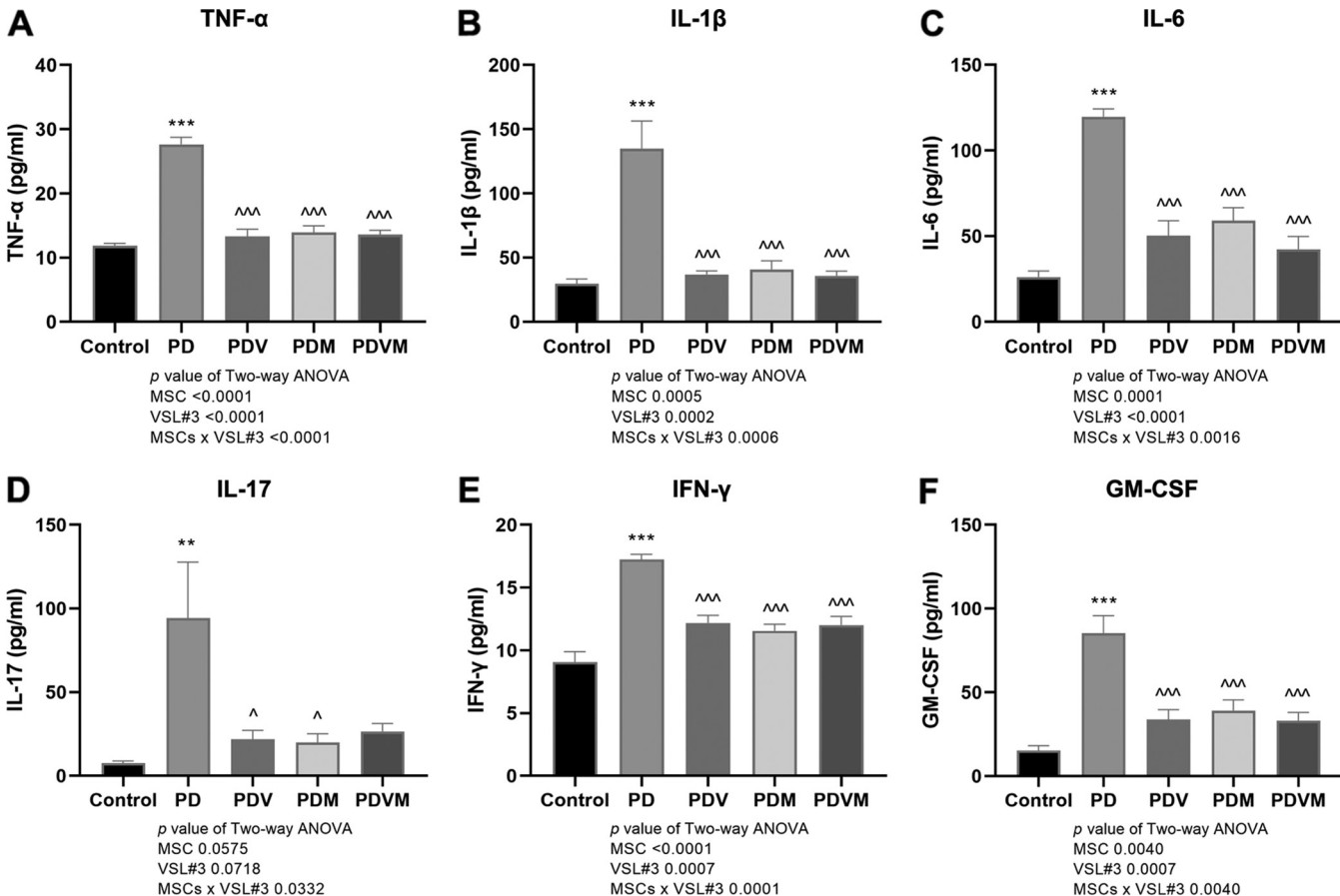

**FIG 4** Effects of VSL#3, hMSCs, and their combination on circulating levels of inflammatory cytokines in PD mice. At euthanasia, the blood was collected and serum was collected after centrifugation. Flow cytometry was performed to quantify the levels of TNF-$\alpha$, IL-1$\beta$, IL-6, IL-17, GM-CSF, and IFN-$\gamma$ in the serum. Data are expressed as means ± SEM ($n = 6$). **, $P < 0.01$; ***, $P < 0.001$ versus control; ^, $P < 0.05$; ^^^, $P < 0.001$ versus PD.

neuroprotection of hMSCs might be through inhibiting the NLRP3 inflammasome. VSL#3, a probiotic formulation currently prescribed for treatment of irritable bowel syndrome, is found to attenuate inflammation and symptoms of PD via restoring the gut microbiota in preclinical models, revealing its potential as an adjuvant approach in PD treatment. Two-way ANOVA indicated that the interactions between VSL#3 and hMSCs in inhibiting central and peripheral inflammation via downregulating the same NLRP3 inflammasome were beneficial to the prevention of dopaminergic neuron loss in PD mice. Surprisingly, neither VSL#3 nor hMSCs alter the anti-inflammatory or neuroprotective activities of the other, suggesting the potential combined use of them for PD treatment. MSCs and probiotic-driven improvement of PD-associated symptoms have been previously reported; however, to the best of our knowledge, the current study was the first that investigated the potential role of the NLRP3 inflammasome in the neuroprotective effects of MSCs and probiotic as well as the potential interactions between them in the PD model.

DA neuronal loss in PD originates from neuroinflammation and is triggered by systemic circulating inflammatory molecules, which can give rise to the release of proinflammatory mediators of neuroinflammation such as TNF-$\alpha$ and IL-1$\beta$ to induce dopaminergic neuron degeneration (23, 24). In accordance with the previous studies, our study showed that together with the degenerative changes in dopaminergic neurons indicated by the dropped dopamine production and the loss of DA neurons, the strikingly elevated expression of proinflammatory cytokines supported the neuroinflammatory status in PD mice. On the other hand, the intraperitoneal injection of MPTP also induces inflammation in peripheral organs, and the inflammation works together with MPTP to interrupt the blood-brain barrier (BBB), causing amplified damage of the nigrostriatal dopaminergic system (6).

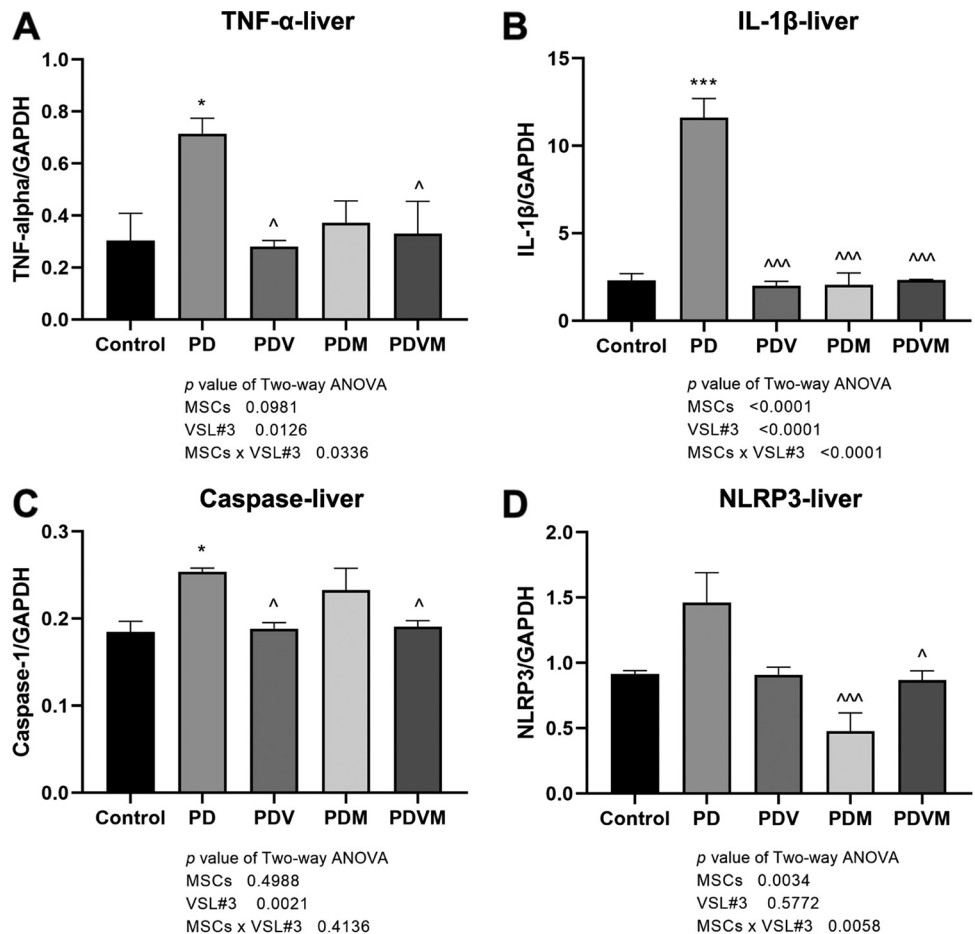

**FIG 5** Effects of VSL#3, hMSCs, and their combination on MPTP-induced mRNA expression of inflammatory cytokines and NLRP3 inflammasome molecules in liver of PD mice. At euthanasia, the liver and intestine were freshly collected. mRNA expression levels of TNF-$\alpha$, IL-1$\beta$, caspase-1, and NLRP3 were measured by real-time PCR. Data are expressed as means $\pm$ SEM ($n = 6$). *, $P < 0.05$; ***, $P < 0.001$ versus control; ^, $P < 0.05$; ^^^, $P < 0.001$ versus PD.

Previous studies reported intestinal inflammation in MPTP-induced mice due to gut microbial dysbiosis and subsequent triggering of the neuroinflammation in the substantia nigra (11, 25). Our present study demonstrated similar inflammatory responses in the intestinal tissues, and the gene expression of proinflammatory cytokines, including TNF-$\alpha$ and IL-1$\beta$, was markedly increased. The liver is an important target of peripheral inflammation, and the impairment of liver functions has also been reported for PD patients, pointing to the connection between brain and liver metabolism (26). Systemic administration of MPTP resulted in inflammatory responses in liver in a dose-dependent manner, as suggested by the upregulated inflammatory markers along with the reduction of striatal dopamine (9). Similarly, notable inflammatory changes took place in the liver in PD mice in this study. Moreover, circulating levels of TNF-$\alpha$ and IL-1$\beta$ outstandingly increased, suggesting the systemic inflammatory deterioration in PD mice to a greater extent. The dramatic elevations were also found in circulating levels of other inflammatory molecules, including IL-6, IL-17, IFN-$\gamma$, and GM-CSF. IL-17 interacted with TNF-$\alpha$, IL-1$\beta$, IFN-$\gamma$, and GM-CSF to exert synergistic activity on inducing inflammatory responses in chronic inflammation. For example, IL-17 together with TNF-$\alpha$ induced the massive release of IL-6, followed by a rapid induction of an extensive range of acute-phase proteins, accumulation of which cause the serious complication of chronic inflammation.

With the increasingly identified importance of central and peripheral inflammation in PD progression, extensive efforts are expanding into anti-inflammatory therapies in PD treatment. Among these therapies, mesenchymal stromal cells and probiotics are two promising approaches. The first appearance of MSCs in the brain happens at 24 h upon intravenous

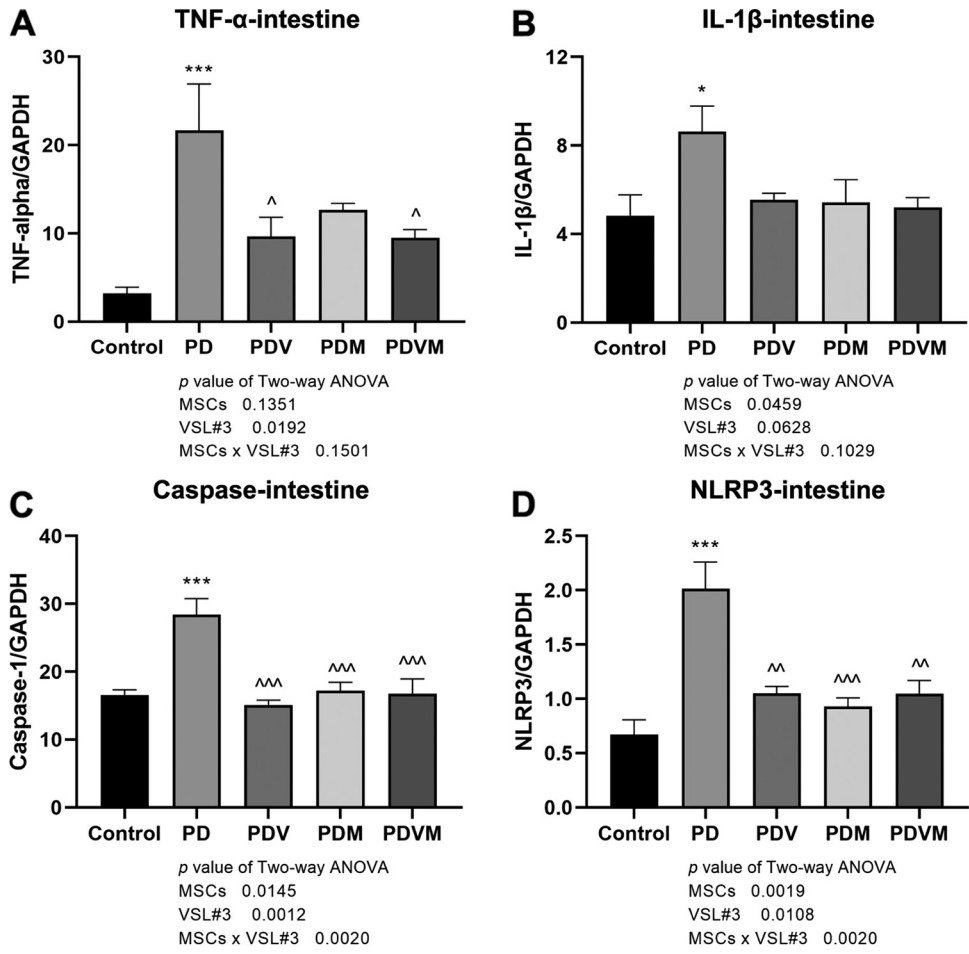

**FIG 6** Effects of VSL#3, hMSCs, and their combination on MPTP-induced mRNA expression of inflammatory cytokines and NLRP3 inflammasome molecules in intestine of PD mice. At euthanasia, the liver and intestine were freshly collected. mRNA expression levels of TNF-$\alpha$, IL-1$\beta$, caspase-1, and NLRP3 were measured by real-time PCR. Data are expressed as means ± SEM ($n = 6$). *, $P < 0.05$; ***, $P < 0.001$ versus control; ^, $P < 0.05$; ^^, $P < 0.01$; ^^^, $P < 0.001$ versus PD.

infusion, with the most powerful signal at day 3 (27), suggesting that MSCs could cross the BBB and create a regenerative microenvironment via the release of bioactive molecules locally (28, 29). In response to 6 consecutive MSC infusions, expression of TNF-$\alpha$ and IL-1$\beta$ was downregulated in the striatum, confirming the local anti-inflammatory effect of MSCs in the brain. This finding also agrees with our previous study with ischemic rats showing that intravenously administered MSCs could enter the brain and exert anti-inflammatory actions (30). However, only a small proportion of MSCs (1 to 2.7% of the transplanted cells) could enter the brain and they only transiently remained there (31), pointing out the crucial role of the systemic mechanisms in mediating the beneficial actions of MSCs in brain. The majority of the intravenously injected MSCs distribute in the peripheral tissues, like tail, blood, lung, liver, kidney, spleen, and gut, up to 10 days after transplantation (27). Indeed, another newly discovered property of MSCs is their potent systemic anti-inflammatory and immunomodulatory potential, which was proven by our results showing that MSCs reduced the serum proinflammatory cytokines. Moreover, MSCs were found to significantly attenuate the inflammatory responses in intestinal tissue and liver to MPTP injection in mice, which is consistent with the published results showing that the respective inhibition of gut and liver inflammation mediates the neuroprotective actions of MSCs in PD models (11, 32).

The NLRP3 inflammasome, a well-characterized sensor molecule, plays a key role in the chronic inflammatory response of PD pathogenesis via caspase-1 activation and consequent secretion of IL-1$\beta$. NLRP3 deficiency prevents the decrease in tyrosine hydroxylase (TH)

expression, loss of the dopaminergic neurons in the substantia nigra, and striatal dopamine production as well as the formation of $\alpha$-synuclein in chronic and subacute MPTP-treated mice (33, 34), indicating that targeting NLRP3, or the crucial molecules of the inflammasome, caspase-1 and IL-1$\beta$, may shed light on PD treatment. Systemic suppression of the NLRP3 inflammasome attenuates motor dysfunction and neuroinflammation in PD mice (35, 36). Human MSCs negatively regulate lipopolysaccharide (LPS)-induced NLRP3 inflammasome activation in macrophages to decrease *in vitro* (37, 38), while the effects of systemic MSC administration on the NLRP3 inflammasome have not been described for PD. Major findings of the present study gave proof of the systemic inhibition by MSCs of NLRP3 and the other two molecules, caspase-1 and IL-1$\beta$, of the inflammasome in the brain (specifically the mid-brain), liver, and gut in PD mice. Regarding the importance of the NLRP3 inflammasome in PD pathogenesis and its therapeutic potential, the findings of our study indicated that MSC infusion might target the NLRP3 inflammasome to exert anti-inflammatory effects in the central system and periphery in PD mice. However, further studies using conditional NLRP3 knockout animals are needed to identify the role of inhibiting the NLRP3 inflammasome in the neuroprotective actions of MSCs.

VSL#3, a probiotic formulation that is currently prescribed for treatment of irritable bowel syndrome, is reported to have neuroprotective effects in the PD model via improving the gut microbial imbalance and attenuating intestinal inflammation. Therefore, VSL#3 has been potentially regarded as an adjuvant approach for PD treatment. Our results confirmed that VSL#3 targeted inflammation in central and peripheral systems to exert neuroprotective effects in PD mice. However, due to the reported regulation of MSCs on gut microbiota and the same anti-inflammatory target, it is of clinical importance to determine the combined efficacy and the potential interactions between VSL#3 and MSCs in terms of PD treatment. Two-way ANOVA demonstrated interactions between VSL#3 and MSCs on inhibiting the anti-inflammatory changes and suppressing the NLRP3 inflammasome in central and peripheral organs to exert a beneficial effect on dopaminergic neurons. Surprisingly, neither them altered the activities of the other with respect to inhibiting inflammation via suppressing the NLRP3 inflammasome and protecting dopaminergic neurons from neurotoxin. Both MSCs and VSL#3 are strong regulators of the immune responses, and the beneficial effects of either of them could be sufficiently potent to be further enhanced by the other at the current dosage, as they share similar targets, including the immune cells, proinflammatory cytokines, and NLRP3 inflammasome, as demonstrated in our study. These results suggest that VSL#3 might be used as an adjuvant approach to MSC infusion for PD treatment. However, further studies to illustrate their combined regulation on gut microbial composition and potential pharmacotoxicity are needed.

As summarized in Fig. 7, the systemic injection of human umbilical cord MSCs (i) inhibited the NLRP3 inflammasome to downregulate the MPTP-induced inflammation in both the central system and periphery and thus (ii) relieved the degenerative changes in dopaminergic neurons in PD mice (Fig. 8). Furthermore, VSL#3 interacts with hMSCs in suppressing the NLRP3 inflammasome, exerting central and peripheral anti-inflammatory effects to exert neuroprotective actions in PD mice, providing evidence for the potential of VSL#3 as an adjuvant therapy in PD treatment. The findings of the present study give a further understanding of the anti-inflammatory activity and the molecular mechanism for the beneficial effects of MSCs as well as the potential application of a probiotic formulation as an adjuvant approach to MSC therapy in PD treatment. However, the effects of MSCs and VSL#3 on inflammation and the NLRP3 inflammasome as well as their interactions over extended periods need to be investigated in future studies.

## MATERIALS AND METHODS

**Isolation, culture, and characterization of human umbilical cord-derived MSCs (hMSCs).** Mesenchymal stromal cells were isolated from human umbilical cord, cultured, and characterized as previously described (21). Briefly, the umbilical cord tissue was weighed and incubated in a 10-cm petri dish with cold Hanks' solution containing penicillin (100 U/mL) and streptomycin (100 $\mu$g/mL). Subsequently, the tissue was cut into three pieces and each piece was thoroughly chopped into 2- to 3-mm fragments in a 50-mL tube with 20 mL of Dulbecco modified Eagle medium (DMEM)-F12 culture medium after removal of the vessels. Supernatant was discarded after centrifugation at 250 $\times$ $g$ and 4°C for 5 min. The cell pellet was resuspended in MSC serum-

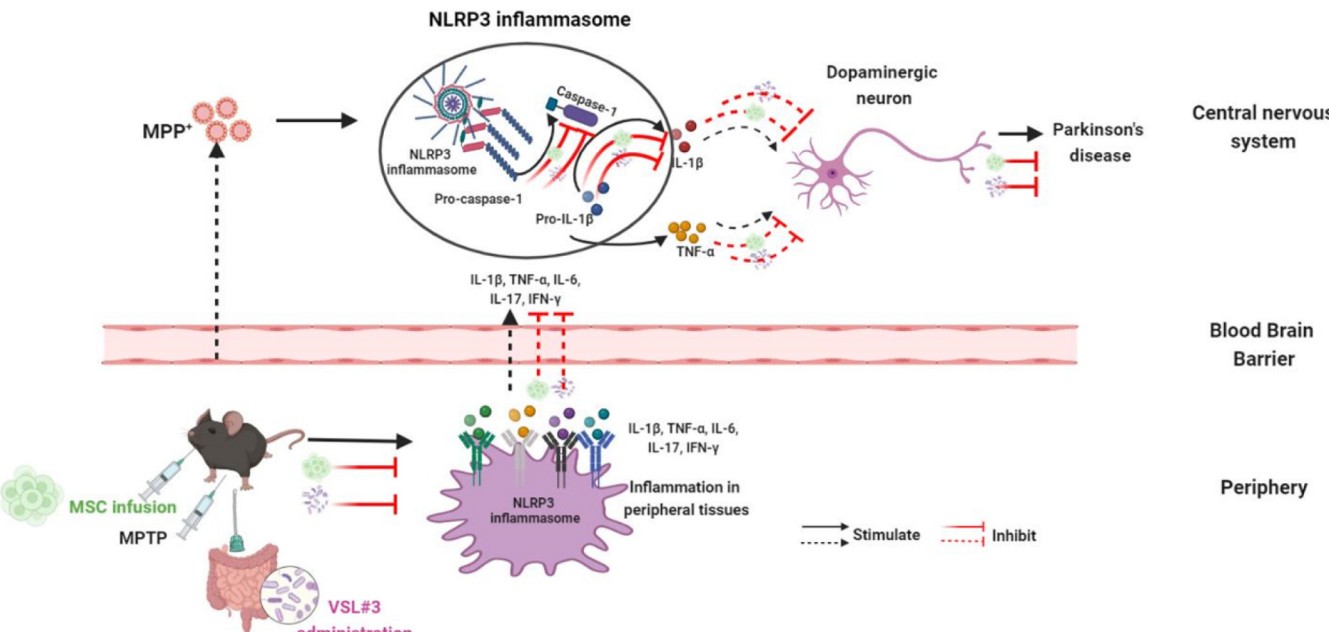

**FIG 7** Possible mechanism for the neuroprotective effects of MSC against MPTP in mice. The systemic injection of human umbilical cord MSCs (i) inhibited the NLRP3 inflammasome to downregulate the MPTP-induced inflammation in both the central system and periphery and thus (ii) relieved the degenerative changes in dopaminergic neurons in PD mice; furthermore, VSL#3 interacts with hMSCs in suppressing NLRP3 inflammasome-mediated inflammatory responses in both the central system and periphery to exert neuroprotective actions in PD mice.

free medium (Yocon, Beijing, China) and cultured in a 37℃ incubator for 4 days. The medium was semichanged to fresh medium, and MSC culture was continued for another 10 days. The tissue fragments were removed and cells (passage 0) were collected upon digestion with stem cell gentle digestive enzyme (Yocon) when the confluence reached 50%. Cells (passage 1) were subcultured at a density of $2 \times 10^6$/T75 flask. The medium was changed every 3 days, and subculture (passage 2) was performed in the same T75 flask when the confluence reached 90%. Cells at passage 3 were used for the characterization and subsequent animal experiments. A FACSCalibur flow cytometer (BD, CA, USA) was used to characterize the immunophenotype of MSCs according to the basic characteristics and minimal criteria of MSCs declared in 2006 by the International Society for Cellular Therapy (ISCT).

**Study design and animal treatment.** All procedures involving animals were performed in accordance with the *Guide for the Care and Use of Laboratory Animals* (39). Seventy-five 10-week-old male C57BL/6 mice were purchased from Beijing Vital River Laboratory Animal Technology Co., Ltd., and housed under a 12-h light-dark cycle with free access to water and food. After 7 days of acclimation, the animals were randomly divided into five groups (15 mice/group): control, PD, PD plus VSL#3 (PDV), PD plus hMSCs (PDM), and PD plus VSL#3 plus hMSCs (PDVM). As shown in Fig. 8, the mice were intraperitoneally injected with MPTP (Sigma; M0896) at 30 mg/kg of body weight/day (PD, PDV, PDM, and PDVM groups) or saline (control group) for 5 consecutive days. From day 7, the mice were weekly administered saline (control and PD groups), VSL#3 (VSL Pharmaceuticals, USA; $4 \times 10^9$ CFU/dose; PDV group) in 0.1 mL of saline via the gastrointestinal tract (PDV group), $2 \times 10^6$ hMSCs in 0.2 mL of 0.9% saline (PDM group) via the tail vein over a 5-min period, and the combination of VSL#3 and hMSCs (PDVM group) for 6 weeks. The dose of VSL#3, dose of hMSCs, and their respective frequencies of administration were selected based on previous studies (18, 30, 40, 41). At euthanasia, blood, brain, striatum, liver, and colon tissue were collected for further measurement.

**HPLC measurement.** The contents of DA and its metabolites, including 3,4-dihydroxyphenylacetic acid (DOPAC), homovanillic acid (HVA), and norepinephrine (NE), were measured using high-performance liquid chromatography (HPLC). Briefly, the left striatum was homogenized in 0.3 mL of liquid A (0.4 M perchloric acid) immediately after weighing. Upon centrifugation at 12,000 rpm for 20 min at 4℃, 120 $\mu$L of the supernatants

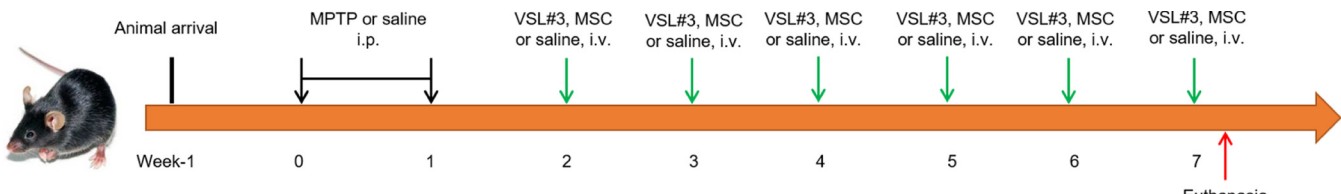

**FIG 8** Workflow of animal experiment. After 1 week of acclimation, the mice were intraperitoneally (i.p.) injected with MPTP at 30 mg/kg/day (PD, PDV, PDM, and PDVM groups) or saline (control group) for 5 consecutive days. From day 7, the mice were weekly administered saline (control and PD groups), VSL#3 (VSL Pharmaceuticals, USA; $4 \times 10^9$ CFU/dose; PDV group) in 0.1 mL of saline via the gastrointestinal tract (PDV group), $2 \times 10^6$ hMSCs in 0.2 mL of 0.9% saline (PDM group) via the tail vein (i.v.) over a 5-min period, and the combination of VSL#3 and hMSCs (PDVM group) for 6 weeks.

was collected and mixed with 60 $\mu$L of liquid B (20 mM citromalic acid-potassium, 300 mM dipotassium phosphate, 2 mM EDTA-2Na). Supernatants (120 $\mu$L) were measured for neurotransmitter content using HPLC assay.

**IHC staining.** Immunohistochemical (IHC) staining was conducted as previously described (42). Briefly, the whole brain was fixed in 4% paraformaldehyde via perfusion. Upon dehydration by sequential soaking in 20% and 30% sucrose solution, 18-$\mu$m serial coronal sections were prepared using a cryostat (Leica, Germany). Upon antigen retrieval, removal of endogenous peroxidases, and blocking of the nonspecific binding with 3% donkey antibody in phosphate-buffered saline (PBS), the slides were incubated with polyclonal rabbit anti-tyrosine hydroxylase (anti-TH) antibody (1:4,000; Sigma-Aldrich, St. Louis, MO, USA) overnight at 4℃, followed by incubation with biotinylated goat anti-rabbit IgG and subsequently with streptavidin peroxidase. TH-positive neurons were visualized by diaminobenzidine (DAB). Images were observed and captured under 100× magnification using a photoscope (BX51; Olympus Corporation, Tokyo, Japan). The average numbers of TH-positive neurons in six sections from each mouse were compared between groups.

**Determination of soluble cytokines.** Blood samples were centrifuged after clotting for 30 min, and serum samples were collected at −80℃. The concentrations of TNF-$\alpha$, IL-1$\beta$, IL-6, IL-17, GM-CSF, and IFN-$\gamma$ in serum were measured using a Legendplex flow-based mouse inflammation panel kit (13-plex) with a V-bottom plate from Biolegend (Fell, Germany) by following the manufacturer's instructions. Briefly, serum samples were 2-fold diluted with the assay buffer. For the standards, 25 $\mu$L of Matrix C was added to the standard wells, followed by 25 $\mu$L of a standard in duplicate. For the samples, 25 $\mu$L of assay buffer was added to the sample wells, followed by 25 $\mu$L of each diluted serum sample in duplicate. Then, 25 $\mu$L of mixed beads was added to each well and shaken at 800 rpm on a plate shaker for 2 h at room temperature. The plate was then centrifuged at 1,050 rpm for 5 min and the supernatant was discarded immediately by quickly inverting and flicking the plate in one continuous and forceful motion. The plate was washed with washing buffer, and the supernatant was discarded as described above, followed by addition of 25 $\mu$L of detection antibodies to each well. The plate was then shaken at 800 rpm on a plate shaker for 1 h at room temperature and 25 $\mu$L of SA-PE was added to each well directly without washing. The plate was incubated on a plate shaker for another 30 min and washed twice, as described above. Washing buffer (150 $\mu$L) was used to resuspend the beads by pipetting. The samples were read by a FACSCalibur flow cytometer (BD, CA, USA). The assay flow cytometry standard (FCS) files were analyzed using Biolegend's LegendPlex data analysis software.

**Real-time PCR assay.** The samples were homogenized in TRIzol reagent by using an electronic homogenizer (Huxi, China). Briefly, 2.0 $\mu$g of total RNA was reverse transcribed into cDNA by using high-capacity cDNA reverse transcription kits (Applied Biosystems) following the manufacturer's instructions. A 20-$\mu$L volume of PCR mixture consisting of 1 $\mu$L of cDNA, 0.4 $\mu$L of forward and reverse primers, 8.2 $\mu$L of DNase- and RNase-free water, and 10 $\mu$L of SsoFast EvaGreen supermix (Bio-Rad) was subjected to PCR using the LightCycler 480 real-time PCR detection system (Roche). Sequences for primers are as follows: TNF-$\alpha$, 3′–5′-CAGGCGGTGCCT ATGTCTC (forward) and 3′–5′-CGATCACCCCGAAGTTCAGTAG (reverse); IL-1$\beta$, 3′–5′-GCCCATCCTCTGTGACT CAT (forward) and AGGCCACAGGTATTTTGTCG (reverse); caspase-1, 3′–5′-ACAAGGCACGGGACCTATG (forward) and 3′–5′-TCCCAGTCAGTCCTGGAAATG (reverse); and NLRP3, 3′–5′-ATGCTGCTTCGACATCTCCT (forward) and 3′–5′-AACCAATGCGAGATCCTGAC (reverse).

**Statistical analysis.** Data were expressed as means ± standard errors of the means (SEM). The Shapiro-Wilk test was performed to determine the normality of the data. The differences between *in vivo* study groups were analyzed by one-way ANOVA with Tukey's *post hoc* test (normal distribution) or Kruskal-Wallis test (non-normal distribution). Interactions between two treatments were analyzed by two-way ANOVA with the Bonferroni test as the *post hoc* test. A P value of <0.05 was considered statistically significant.

**Ethics approval.** The animal experiments were approved by the Xuanwu Hospital Capital Medical University Animal Subjects Ethics Sub-committee (reference no. 21-07-21).

**Data availability.** Data are available at https://figshare.com/s/b4a2bcb6dedcb225c6a6.

## ACKNOWLEDGMENTS

We thank Xueying Song at Central Laboratory of Capital Medical University for the HPLC performance.

We have no competing interests to declare.

Liping Zhou performed the animal experiment, sample collection and measurement, and data analysis and wrote the manuscript. Deqiang Han and Xingzhe Wang helped with the animal experiment and sample collection. Zhiguo Chen conceived and supervised the study and finalized the manuscript. All authors reviewed the manuscript and approved the submission.

This work was supported by the Stem Cell and Translation National Key Project (2016YFA0101403), National Natural Science Foundation of China (82171250 and 81973351), Beijing Municipal Health Commission Fund (PXM2020_026283_000005) to Zhiguo Chen, and Beijing Postdoctoral Research Foundation (2021-ZZ-005) to Liping Zhou.

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
