## [Reviewer comments · Microbiology Spectrum]

Microbiology Spectrum

Probiotic formulation VSL#3 interacts with mesenchymal stromal cells to protect dopaminergic neurons via centrally and peripherally suppressing NLRP3 inflammasome-mediated inflammation in Parkinson's disease mice

Liping Zhou, Deqiang Han, Xingzhe Wang, and Zhiguo Chen

Corresponding Author(s): Zhiguo Chen, Xuanwu Hospital Capital Medical University

Review Timeline:

Submission Date:	August 18, 2022
Editorial Decision:	November 29, 2022
Revision Received:	December 3, 2022
Accepted:	January 8, 2023

Editor: Po-Yu Liu

Reviewer(s): Disclosure of reviewer identity is with reference to reviewer comments included in decision letter(s). The following individuals involved in review of your submission have agreed to reveal their identity: Yi-Ju Chen (Reviewer #3)

Transaction Report:

DOI: <https://doi.org/10.1128/spectrum.03208-22>

November 29, 2022

Prof. Zhiguo Chen
Xuanwu Hospital Capital Medical Univeristy
Cell Therapy Center
Beijing
China

Re: Spectrum03208-22 (Probiotic formulation VSL#3 interacts with mesenchymal stromal cells to protect dopaminergic neurons via centrally and peripherally suppressing NLRP3 inflammasome-mediated inflammation in Parkinson's disease mice)

Dear Prof. Zhiguo Chen:

Link Not Available

Sincerely,

Po-Yu Liu

Journals Department
Reviewer comments:

Reviewer #1 (Comments for the Author):

This study reports the neuroprotective activities of human umbilical cord MSCs from the aspect of anti-inflammation actions. The authors showed that hMSCs inhibit the NLRP3 inflammasome and the MPTP-induced inflammation in the brain and periphery to relieve the degenerative changes in dopaminergic neurons in a model of PD in mice. In addition, the authors combined probiotic formulations and found that VSL#3 interacts with hMSCs on suppressing NLRP3 inflammasome to exert neuroprotective actions in the mice model of PD. The combination did not alter the actions of hMSCs, and based on this finding, the authors suggested that VSL#3 might have potential as adjuvant therapy in PD treatment. The findings of the present study could push the field forward in the understanding of the molecular mechanism for the anti-inflammatory action of MSCs and probiotics in PD. The

authors are encouraged to address the following points:

Only male animals were used. Based on the ARRIVE guidelines, both sexes are strongly recommended to be employed in basic animal studies. Please explain why females were not used.

Please add the total number of animals used in this study in the method section.

Please explain why i.v. the administration was used, and if other routes of administration can be used. Most probiotics are used after oral administration.

In the statistic section, please provide further information about the statistical package used, normality tests, and also the interaction analysis. It seems that the authors have used parametric tests. does this mean that the data were normally distributed? which test was used? Please add the result of checking.

In figures of interaction, please clarify the interaction between which and what. This can be in the figure legend or text related.

In the proposed graphical schematic, both green and red are used for inhibition. It can be confusing as green can be considered promoting. can authors exchange the way for inhibition from the two ways with the aid of another mode or type?

What are the limitations of this study?

The authors have the effect of hMSCs and probiotics alone and in conjugation. Can the authors discuss whether the net effect is additive or synergistic?

What are the potential side effects or unwanted issues occurring in these treatments? Did the authors include a safety indicator in their outcome measures?

In this study the authors have investigated the molecular mechanisms, but what about the behavioral tests and the final overall effect on PD in the mice model? motor or sensory or both? cognitive, or non-cognitive? If the behavioral tests are not performed, add literature or speculation assumptions.

What are the translational challenges if the findings from this study are to be tested in humans? What about the

Please add how the dose and time of administration were decided. This is important if the administration concentration and time can influence the outcome (i.e., if the effects seen are dose or time-dependent)

Reviewer #3 (Comments for the Author):

The authors reported the potential effect of probiotic VSL-3# and human mesenchymal stromal cells in mice models of Parkinson's disease. Both treatments protected dopaminergic neuron loss and exerted systemic anti-inflammatory effects both in the neuron and GI tract. The manuscript is well-written, and the study is carefully conducted. Some questions may need to be answered.

1. In most experiments, both VSL-3# and hMSCS exert a similar effect on the anti-inflammatory function or neuron protection. The combination of both treatments may exert a more beneficial effect. Please explain and discuss this.
2. Did the authors investigate the role of hMSC in the GI tract? Since it also provided an anti-inflammation effect on the intestinal cells. How about the role in intestinal permeability or gut microbiome modification?
3. Likewise, did the authors investigate the effect of VSL-3# in the GI tract regarding the gut microbiota or intestinal permeability in PD mice?

Staff Comments:

Preparing Revision Guidelines

Please return the manuscript within 60 days; if you cannot complete the modification within this time period, please contact me. If you do not wish to modify the manuscript and prefer to submit it to another journal, please notify me of your decision immediately so that the manuscript may be formally withdrawn from consideration by Microbiology Spectrum.

Reviewer comments:

Reviewer #1 (Comments for the Author):

This study reports the neuroprotective activities of human umbilical cord MSCs from the aspect of anti-inflammation actions. The authors showed that hMSCs inhibit the NLRP3 inflammasome and the MPTP-induced inflammation in the brain and periphery to relieve the degenerative changes in dopaminergic neurons in a model of PD in mice. In addition, the authors combined probiotic formulations and found that VSL#3 interacts with hMSCs on suppressing NLRP3 inflammasome to exert neuroprotective actions in the mice model of PD. The combination did not alter the actions of hMSCs, and based on this finding, the authors suggested that VSL#3 might have potential as adjuvant therapy in PD treatment. The findings of the present study could push the field forward in the understanding of the molecular mechanism for the anti-inflammatory action of MSCs and probiotics in PD. The authors are encouraged to address the following points:

1. *Only male animals were used. Based on the ARRIVE guidelines, both sexes are strongly recommended to be employed in basic animal studies. Please explain why females were not used.*

Response: Thanks for the comment. We totally agree with the reviewer that both sexes should be included in conducting the basic animal studies. There are two main reasons why only male animals were employed in the present study:

1) According to the epidemiological and clinical features of the disease, PD affects men twice more often than women (Cerri *et al.*, 2019). Moreover, the male to female ratios of PD incidence increase with age (Moisan *et al.*, 2016). Therefore, the mature male animals were used to mimic the higher risk of developing PD in male population in the aspect of epidemiology;

2) The female animals exhibit a higher level of estradiol and estradiol that work as a potent anti-inflammation agent and can downregulate pro-inflammatory molecules (Villa *et al.*, 2015) and interact with the gut microbiota (Lephart *et al.*, 2022; Song *et al.*, 2020). This effect might complicate the anti-inflammatory activities of MSCs and probiotic. And the inflammatory response within female has also been shown to vary significantly according to the estrous cycle in rodents, making the anti-inflammatory study on MSCs and probiotic more complicated. Therefore, we did not use female animals in the present study.

In our future work, we may propose a new experiment to specifically investigate the interaction or effects of estrogen on the beneficial effects of MSCs.

References

Cerri, Silvia, Liudmila Mus, and Fabio Blandini. "Parkinson's disease in women and men: what's the difference?." *Journal of Parkinson's disease* 9.3 (2019): 501-515.

Moisan, Frédéric, et al. "Parkinson disease male-to-female ratios increase with age: French nationwide study and meta-analysis." *Journal of Neurology, Neurosurgery & Psychiatry* 87.9 (2016): 952-957.

Villa, Alessandro, et al. "Estrogen accelerates the resolution of inflammation in macrophagic cells." *Scientific reports* 5.1 (2015): 1-14.

Lephart, Edwin D., and Frederick Naftolin. "Estrogen action and gut microbiome metabolism in dermal health." *Dermatology and Therapy* (2022): 1-16.

Song, Chin-Hee, et al. "17 β -Estradiol supplementation changes gut microbiota diversity in intact and colorectal cancer-induced ICR male mice." *Scientific reports* 10.1 (2020): 1-14.

2. *Please add the total number of animals used in this study in the method section.*

Response: Thank you for pointing this out. The total number of animals has been provided and highlighted in the revised manuscript. Please kindly refer to line 412.

3. *Please explain why i.v. the administration was used, and if other routes of administration can be used. Most probiotics are used after oral administration.*

Response: Thanks for the comment. Yes, in our study, MSCs were given to animals via i.v. injection while probiotic was given via gastrointestinal tract. For MSCs, several methods of administration are often used, including local site transplantation, intravenous injection, intranasal administration. As we focused on the immune-based anti-inflammatory effects of MSCs in both the central and peripheral tissues, MSCs were intravenously injected via tail vein. The probiotic was given to the animal via gastrointestinal administration to mimic the oral administration in human. Please kindly refer to the detail of the treatment (lines 418 to 421) in the methods part of our manuscript.

4. *In the statistic section, please provide further information about the statistical package used, normality tests, and also the interaction analysis. It seems that the authors have used parametric tests. does this mean that the data were normally distributed? which test was used? Please add the result of checking.*

Response: We thank the reviewer for the comments. One-way ANOVA with Tukey's post hoc test (normal distribution) or Kruskal-Wallis test (non-normal distribution) was performed to determine the difference between groups, while Two-way ANOVA with Bonferroni as post hoc test was performed to determine the possible interaction between MSCs and probiotic formulation VSL#3 for all the parameters by using GraphPad Prism 0.0.0. Before that, Shapiro-Wilk test was performed to see whether the data was normally distributed as our sample size of the present study is relatively small. Shapiro-Wilk test is a hypothesis test that is applied to a sample and whose null hypothesis is that the sample has been generated from a normal population. If the *p* value is <0.05, we reject such a null hypothesis and say that the sample has not been generated from a normal distribution. The detail of the statistical analysis has been provided in the revised manuscript. Please kindly refer to lines 472 to 475.

5. *In figures of interaction, please clarify the interaction between which and what. This can be in the figure legend or text related.*

Response: We thank the reviewer for the comments and apologize for the unclear data presentation. The interaction between hMSCs and VSL#3 (VSL#3 \times hMSCs) has

been indicated in the corresponding text. Please kindly refer to the revised manuscript.

6. *In the proposed graphical schematic, both green and red are used for inhibition. It can be confusing as green can be considered promoting. can authors exchange the way for inhibition from the two ways with the aid of another mode or type?*

Response: We apologize for the confusing presentation. The graphical abstract has been revised in the updated figures. Please kindly refer to the updated figure 7.

7. *What are the limitations of this study?*

Response: The treatment with either MSCs or VSL#3 lasted for only several weeks in the present study. As their anti-inflammatory effects and potential interaction between them might be correlated with the treatment period, therefore, the effects of MSCs and VSL#3 on inflammation and NLRP3 inflammasome as well as their interactions over extended periods need to be investigated in future studies. The limitation has been added in the discussion part of the revised manuscript. Please refer to lines 387 to 389.

8. *The authors have the effect of hMSCs and probiotics alone and in conjugation. Can the authors discuss whether the net effect is additive or synergistic?*

Response: Thanks for the comment. Both the additive and synergistic effects are used to define the interactions between two or more substances, usually chemicals. For additive effect, it usually occurs with chemicals that are similar in structure, so they work as a team. Their combined effects are the sum of their respective effects, for example, $2+2=4$. For the synergistic effects, the substances are designed to work well on its own and they may create dangerous situation when they are used together because their combined powers are thought to be overwhelming. In the present study, neither MSCs nor VSL#3 are chemical and do not share the same structure, but both of them exert potent inhibitory effects on inflammation in both central and peripheral tissues via the same targets as demonstrated in our study. In the present study, the combined effects of MSCs and VSL#3 appeared not to be stronger than each of them individually possibly because their respective effects were too potent to be enhanced by the other at the current dose. It is speculated that the interactions between them might be “additive effects” at certain dosages, which needs to be further studied.

9. *What are the potential side effects or unwanted issues occurring in these treatments? Did the authors include a safety indicator in their outcome measures?*

Response: We appreciate the comment of the reviewer very much. Yes, there should be evaluation of the potential side effects upon treatments with MSCs and probiotic, especially when they are used together, like the commonly measured pharmacological toxicities in kidney, liver, spleen and others. Actually, as MSCs are different from the pharmacological agents in components and administration, they should have some specific evaluations regarding their safety, like the formation of tumor and thrombus. In fact, we have been doing literature reviewing to determine the specific parameters for evaluating the side effects of MSCs in addition to the pharmacological toxicities.

We appreciate if the reviewer could give some suggestion on this kind of study.

10. *In this study the authors have investigated the molecular mechanisms, but what about the behavioral tests and the final overall effect on PD in the mice model? motor or sensory or both? cognitive, or non-cognitive? If the behavioral tests are not performed, add literature or speculation assumptions.*

Response: Thanks for the comment. The beneficial effects of MSCs in PD treatment have been widely studied. The improvement of umbilical cord MSCs on alleviating the locomotor deficits has been previously reported in the same MPTP-induced PD mice (Sun *et al.*, 2022). Based on these findings, we mainly focused on the mechanistic investigation in the present study in the aspect of anti-inflammation and related pathways. The literature has been added in the introduction part of the revised manuscript. Please kindly refer to lines 104 to 105.

References

Sun, Zhengqin, et al. "Human umbilical cord mesenchymal stem cells improve locomotor function in Parkinson's disease mouse model through regulating intestinal microorganisms." *Frontiers in Cell and Developmental Biology* (2022): 3740.

11. *What are the translational challenges if the findings from this study are to be tested in humans?*

Response: Thanks for the comment. Actually, the effects of MSCs against PD have already been reported in human, providing promising preliminary data. Currently, according to ClinicalTrial.gov, 13 clinical trials worldwide are in progress investigating the safety and efficacy of MSCs in PD patients with highly variable setup, lacking consistency between trials. Therefore, these studies are difficult to compare with each other, which might be the biggest challenge for translation. Besides, the sources of variability in MSCs-based PD treatment also need to be standardized.

12. *Please add how the dose and time of administration were decided. This is important if the administration concentration and time can influence the outcome (i.e., if the effects seen are dose or time-dependent)*

Response: Thanks for the comment. The dosage of VSL#3 in mice were selected based on the published studies in mice (Mariman *et al.*, 2015; Chen *et al.*, 2019; Theriot *et al.*, 2022). The frequency of administration was decided based on the finding that major differences in the gut microbiota were seen in the mice between control and VSL#3 treatments in feces over time and in the ceca of mice at day 7 (Theriot *et al.*, 2022), which suggest that the probiotic strains took several days to drive changes in the microbial community structure. The number of MSCs injected was selected based on the previous study conducted in rats from our team (Guan *et al.*, 2021) and the frequency of treatment was decided based on the fact that it takes 7 to 10 days for the intravenously injected MSCs to migrate into major tissues all around the body (Kraitchman *et al.*, 2005). All these studies have been cited in our

manuscript. The detailed selection was provided in the method part (lines 422 to 423). As stated in response to reviewer's question 7, the anti-inflammatory effects, especially the potential interaction between MSCs and VSL#3 might somehow be correlated with the period of treatment. Therefore, their respective effects and potential interactions on inflammation responses and NLRP3 inflammasome in both central and peripheral tissues as well as their protective effects on dopaminergic neurons over extended period will be proposed in our future studies.

References

- Mariman, Rob, et al. "The probiotic mixture VSL# 3 has differential effects on intestinal immune parameters in healthy female BALB/c and C57BL/6 mice." *The Journal of nutrition* 145.6 (2015): 1354-1361.
- Chen, Xiaohong, et al. "Bifidobacterium longum and VSL# 3® amelioration of TNBS-induced colitis associated with reduced HMGB1 and epithelial barrier impairment." *Developmental & Comparative Immunology* 92 (2019): 77-86.
- Theriot, Casey, et al. "Probiotic colonization dynamics after oral consumption of VSL# 3® by antibiotic-treated mice." *Microbiome Research Reports* 1.4 (2022): 21.
- Guan, Yunqian, et al. "Astrocytes constitute the major TNF- α -producing cell population in the infarct cortex in dMCAO rats receiving intravenous MSC infusion." *Biomedicine & Pharmacotherapy* 142 (2021): 111971.
- Kraitzman, Dara L., et al. "Dynamic imaging of allogeneic mesenchymal stem cells trafficking to myocardial infarction." *Circulation* 112.10 (2005): 1451-1461.

Reviewer #3 (Comments for the Author):

The authors reported the potential effect of probiotic VSL-3# and human mesenchymal stromal cells in mice models of Parkinson's disease. Both treatments protected dopaminergic neuron loss and exerted systemic anti-inflammatory effects both in the neuron and GI tract. The manuscript is well-written, and the study is carefully conducted. Some questions may need to be answered.

1. *In most experiments, both VSL-3# and hMCSL exert a similar effect on the anti-inflammatory function or neuron protection. The combination of both treatments may exert a more beneficial effect. Please explain and discuss this.*

Response: We thank reviewer 3 for the comment. There are several types of effects to describe the interactions, usually between chemicals, including the additive effects, synergistic effects and antagonistic effects. In our case, the additive and synergistic effects are suitable to define the interactions between MSCs and VSL#3. For additive effect, it usually occurs with chemicals that are similar in structure, so they work as a team. Their combined effects are the sum of their respective effects, for example, 2+2=4. For the synergistic effects, the substances are designed to work well on its own and they may create dangerous situation when they are used together because their combined powers are thought to be overwhelming and the effects of their

combination will be greater than that of either substance taken separately. That's why it is of importance to study the interactions between two substances having similar or same target. However, in the present study, neither MSCs nor VSL#3 add more to the effect of each other on inhibiting the inflammatory responses and NLRP3 inflammasome, protecting dopaminergic neurons possibly because the beneficial effects of either them were already too potent to be enhanced by the other one sharing the similar targets, like the immune cells, the pro-inflammatory cytokines, and NLRP3 inflammasome as demonstrated in our study. The discussion has been updated and please kindly refer to line 366 to 373 in our revised manuscript.

2. *Did the authors investigate the role of hMSC in the GI tract? Since it also provided an anti-inflammation effect on the intestinal cells. How about the role in intestinal permeability or gut microbiome modification?*

Response: Thanks for the comment. For the role of hMSCs in the GI tract, we measured the influences of hMSCs on the inflammatory responses in the intestinal tissues and found that hMSCs exerted significant inhibitory effects on the pro-inflammatory cytokines, including TNF- α and NLRP3 inflammasome molecules. However, we did not measure the intestinal permeability or gut microbiota composition in this study as the target endpoint of intestinal permeability and gut microbial composition are the immune responses. Moreover, the regulation of MSCs on gut microbial composition and intestinal permeability have been reported in various models (Zhao *et al.*, 2021; Ocansey *et al.*, 2019; Yan *et al.*, 2018). Nevertheless, we agree with the reviewer and will try to clarify the role of hMSCs in intestinal permeability and gut microbial composition by using PD animals.

References

- Zhao, Lin-Na, et al. "Bone marrow mesenchymal stem cell therapy regulates gut microbiota to improve post-stroke neurological function recovery in rats." *World Journal of Stem Cells* 13.12 (2021): 1905.
- Ocansey, Dickson Kofi Wiredu, et al. "Mesenchymal stem cell–gut microbiota interaction in the repair of inflammatory bowel disease: an enhanced therapeutic effect." *Clinical and Translational Medicine* 8.1 (2019): 1-17.
- Yan, Nannan, et al. "Human umbilical cord-derived mesenchymal stem cells ameliorate the enteropathy of food allergies in mice." *Experimental and Therapeutic Medicine* 16.6 (2018): 4445-4456.

3. *Likewise, did the authors investigate the effect of VSL-3# in the GI tract regarding the gut microbiota or intestinal permeability in PD mice?*

Response: Thanks for the comment. VSL#3 is a commercial probiotic mixture consisting of eight bacterial strains with the aim to restore the imbalance in gut microbial composition in dysbiosis. Study illustrated the ability of VSL#3 probiotics to colonize the host and impact the gut microbiota composition (Theriot *et al.*, 2022). These colonized strains have been demonstrated to reduce the intestinal permeability in patients with irritable bowel syndrome (Boonma *et al.*, 2021) and animals (Cruz *et al.*, 2021). Based on these findings, we did not measure the influence of VSL#3 on the gut microbial composition or intestinal permeability in the present study.

References

Theriot, Casey, et al. "Probiotic colonization dynamics after oral consumption of VSL# 3® by antibiotic-treated mice." *Microbiome Research Reports* 1.4 (2022): 21.

Boonma, Prapaporn, et al. "Probiotic VSL# 3 treatment reduces colonic permeability and abdominal pain symptoms in patients with irritable bowel syndrome." *Frontiers in Pain Research* (2021): 66.

Cruz, Bruna Cristina dos Santos, et al. "Evaluation of the efficacy of probiotic VSL# 3 and synbiotic VSL# 3 and yacon-based product in reducing oxidative stress and intestinal permeability in mice induced to colorectal carcinogenesis." *Journal of Food Science* 86.4 (2021): 1448-1462.

January 8, 2023

Prof. Zhiguo Chen
Xuanwu Hospital Capital Medical Univeristy
Cell Therapy Center
Beijing
China

Re: Spectrum03208-22R1 (Probiotic formulation VSL#3 interacts with mesenchymal stromal cells to protect dopaminergic neurons via centrally and peripherally suppressing NLRP3 inflammasome-mediated inflammation in Parkinson's disease mice)

Dear Prof. Zhiguo Chen:

Your manuscript has been accepted, and I am forwarding it to the ASM Journals Department for publication. You will be notified when your proofs are ready to be viewed.

Sincerely,

Po-Yu Liu
Editor, Microbiology Spectrum
